# Towards Making the Most of ChatGPT for Machine Translation

**Keqin Peng**$^{\diamond,\Re}$*, **Liang Ding**$^{\Re}$†, **Qihuang Zhong**$^{\sharp}$, **Li Shen**$^{\Re}$
**Xuebo Liu**$^{\flat}$, **Min Zhang**$^{\flat}$, **Yuanxin Ouyang**$^{\diamond}$, **Dacheng Tao**$^{\heartsuit}$
$^{\diamond}$Beihang University $^{\Re}$JD Explore Academy $^{\sharp}$Wuhan University
$^{\heartsuit}$The University of Sydney $^{\flat}$Harbin Institute of Technology, Shenzhen
✉ keqin.peng@buaa.edu.cn, liangding.liam@gmail.com
🧰 https://github.com/Romainpkq/ChatGPT4MT

## Abstract

ChatGPT shows remarkable capabilities for machine translation (MT). Several prior studies have shown that it achieves comparable results to commercial systems for high-resource languages, but lags behind in complex tasks, e.g., low-resource and distant-language-pairs translation. However, **they usually adopt simple prompts which can not fully elicit the capability of ChatGPT**. In this paper, we aim to further mine ChatGPT's translation ability by revisiting several aspects: 🌡 temperature, 🎯 task information, and 🕐 domain information, and correspondingly propose an optimal temperature setting and two *(simple but effective)* prompts: *Task-Specific Prompts* (TSP) and *Domain-Specific Prompts* (DSP). We show that: ❶ The performance of ChatGPT depends largely on temperature, and a lower temperature usually can achieve better performance; ❷ Emphasizing the task information can further improve ChatGPT's performance, particularly in complex MT tasks; ❸ Introducing domain information can elicit ChatGPT's generalization ability and improve its performance in the specific domain; ❹ ChatGPT tends to generate hallucinations for non-English-centric MT tasks, which can be partially addressed by our proposed prompts but still need to be highlighted for the MT/NLP community. We also explore the effects of advanced in-context learning strategies and find a *(negative but interesting)* observation: the powerful chain-of-thought prompt leads to word-by-word translation behavior, thus bringing significant translation degradation.

## 1 Introduction

Recently, the emergence of ChatGPT[1] has brought remarkable influence on natural language processing (NLP) tasks. ChatGPT is a large-scale language model developed by OpenAI, based on Instruct-GPT (Ouyang et al., 2022a), that has been trained to follow instructions with human feedback. ChatGPT possesses diverse abilities of NLP, including question answering, dialogue generation, code debugging, generation evaluation, and so on (Qin et al., 2023; Zhong et al., 2023; Wang et al., 2023a; Kocmi and Federmann, 2023; Lu et al., 2023b; Wang et al., 2023b). We are particularly interested in how well ChatGPT can perform on the machine translation task.

Previous studies (Jiao et al., 2023; Hendy et al., 2023) on translation tasks have found that ChatGPT performs competitively with commercial translation products (e.g., Google Translate and Microsoft Translator) on high-resource languages, but has limited capabilities for low-resource and distant languages. However, they only adopt simple prompts and basic settings regardless of the significant influence of the prompts' quality (Zhou et al., 2022), which may limit ChatGPT's performance. In this paper, we aim to further elicit the capability of ChatGPT by revisiting the following three aspects and correspondingly propose an optimal temperature setting and two simple but effective prompts: *Task-Specific Prompts* (TSP) and *Domain-Specific Prompts* (DSP).

🌡 **Temperature.** Temperature is an important parameter to ensure ChatGPT generates varied responses to human queries. Basically, decoding with higher temperatures displays greater linguistic variety, while the low one generates grammatically correct and deterministic text (Ippolito et al., 2019). However, for tasks with a high degree of certainty, such as machine translation, we argue, a diverse generation may impede its translation quality. We evaluate the performance of ChatGPT at different temperatures to verify its effect and find the optimal temperature setting for the following experiments.

---

*Work was done when Keqin was interning at JD Explore Academy.

†Corresponding Author.

[1]https://chat.openai.com

🎯 **Task Information.** ChatGPT is fine-tuned on high-quality chat datasets and thus essentially a conversational system that has a certain distance from the translation system, we argue that the task inconsistency will limit its translation ability to a certain degree. In response to this problem, we proposed *Task-Specific Prompts* (TSP) to further emphasize the task information to bridge the task gap, i.e., conversation and translation.

🧭 **Domain Information.** Compared with traditional machine translation systems, ChatGPT can incorporate additional information, like human interactions, through the input prompts (Dong et al., 2023). We argue that such flexible interaction may alleviate some classical MT challenges, e.g., cross-domain generalization (Koehn and Knowles, 2017). We, therefore, propose *Domain-Specific Prompts* (DSP) to introduce the domain navigation information to elicit ChatGPT's generalization ability across different domains.

Through extensive experiments, we find that:

📌 ChatGPT's performance largely depends on the temperatures, especially in difficult languages. Generally, setting a lower temperature can result in higher performance.

📌 Emphasizing the task information in prompts can further improve ChatGPT's performance, especially in complex tasks.

📌 Introducing the correct domain information consistently improves ChatGPT's performance while wrong domain information leads to significant degradation in performance.

🚨 When tackling the non-English-centric tasks (both the input and expected output are non-English), ChatGPT may generate hallucinations, which should be paid more attention to by the MT/NLP community.

Furthermore, we explore the effects of several advanced in-context learning strategies (Brown et al., 2020b). Specifically, we investigate ChatGPT's few-shot in-context learning (ICL) and chain-of-thought (CoT) (Wei et al., 2022c; Kojima et al., 2022) abilities on MT tasks. Experimental results show that few-shot ICL can further improve ChatGPT's performance, which is identical to the findings of Hendy et al. (2023), and we also find a negative but interesting observation: CoT leads to

| Test Set | Direction | Domain | Size |
|---|---|---|---|
| Flores-200 | Any | General | 1,012 |
| WMT19 News | En⇒Zh | News | 2,001 |
| | En⇒De | | 3,004 |
| WMT19 Bio | En⇒Zh | Biomedical | 224 |
| | Zh⇒En | | 241 |
| WMT22 E-Commerce | En⇒Zh | E-Commerce | 530 |

Table 1: Data statistics and descriptions.

word-by-word translation behavior, thus bringing significant translation degradation. Also, we call for improving ICL and CoT for MT upon ChatGPT by incorporating the philosophy of example-based and statistical MT (Nagao, 1984; Koehn, 2009).

The remainder of this paper is designed as follows. We present the evaluation settings in Section 2. In Section 3, we revisit the performance of ChatGPT from three aspects (temperature, task, and domain information) and show the zero-shot translation performance of ChatGPT with our proposed advanced prompt recipes. Section 4 summarizes the few-shot in-context learning and chain-of-thought results. Section 6 presents conclusions.

## 2 Evaluation Setting

We provide a brief introduction of the evaluation setting, which mainly includes the used models, test set, and evaluation metrics.

**Models.** We mainly compare ChatGPT[2] with the commercial translation product Google Translator[3], which supports translation in 133 languages. By default, the results in this paper come from the *gpt-3.5-turbo-0301* models, which power the ChatGPT.

**Data.** For multilingual translation and in-context learning, we evaluate the performance of the models on the Flores-200 (Goyal et al., 2022)[4] test sets, which consists of 1012 sentences translated into 204 languages. To evaluate the effect of cross-domain translation, we adopt the test set of WMT19 Biomedical (Bawden et al., 2019), News Translation Task (Barrault et al., 2019) and WMT22 E-Commerce task (Kocmi et al., 2022). Table 1 lists the statistics of these test sets. We test all samples through OpenAI API.

**Metric.** The translation metrics shared task (Freitag et al., 2022) recommends using neural network-

[2] https://chat.openai.com/chat
[3] https://translate.google.com
[4] https://github.com/facebookresearch/flores

| Method | Translation Prompt |
|---|---|
| **ChatGPT** | `"role": "user", "content": "Please provide the [TGT] translation for the following sentence:"` |
| **ChatGPT + TSP** | `"role": "system", "content": "You are a machine translation system.", "role": "user", "content": "Please provide the [TGT] translation for the following sentence:"` |

Table 2: Multilingual translation prompts.

based metrics since they have demonstrated a high correlation with human evaluation and are resilient to domain shift. Hence, we adopt the mostly used **COMET** (Rei et al., 2020) as our primary metric and use the default parameters of "comet-compare" for significance test[5]. Specifically, we use the reference-based metric COMET-20 (*wmt20-COMET-da*). Additionally, we also report BLEU scores (Papineni et al., 2002) and **ChrF** (Popović, 2015) using **SacreBLEU** (Post, 2018) for completeness, but notably, we mainly analyze the performance in terms of model-based metric COMET.

## 3 Zero-Shot Translation

In this section, we explore the performance of Chat-GPT from three aspects: TEMPERATURE, TASK INFORMATION, and DOMAIN INFORMATION, and correspondingly propose an optimal temperature setting and two simple and effective prompts to improve ChatGPT's performance.

### 3.1 The Effect of Temperature 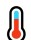

ChatGPT is a chatting machine designed to provide fluent and diverse responses to a wide range of human requests. It is intuitive that the diversity of responses may hinder its performance on tasks with a high degree of certainty, such as machine translation, to some extent.

To investigate the influence of diversity, we compare the performance of ChatGPT in different temperature settings, including 0, 0.2, 0.4, 0.6, 0.8, and 1, across three translation directions: English⇒Romanian, English⇒Chinese, and English⇒German. The relationship between temperature and performance of ChatGPT is shown in Figure 1 and 2.

**Results.** Figure 1 and 2 show that ChatGPT's performance largely depends on the value of temperatures, and as the temperature rises, there is

[5] https://github.com/Unbabel/COMET

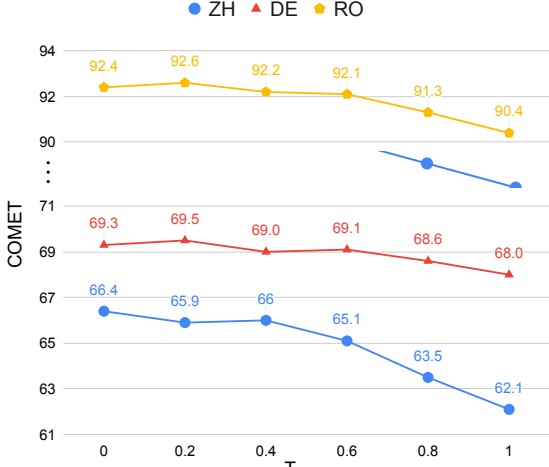

Figure 1: The relationship between temperature and ChatGPT's performance (in terms of COMET scores) when translating from English to other languages.

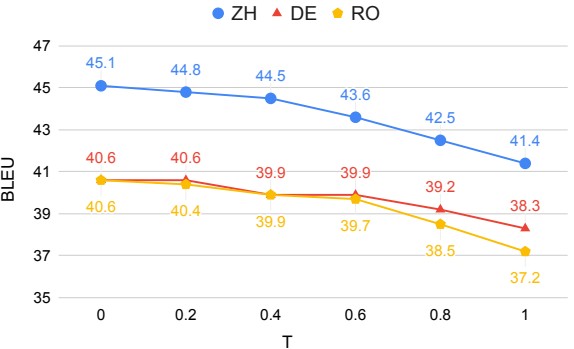

Figure 2: The relationship between temperature and ChatGPT's performance (in terms of BLEU scores) when translating from English to other languages.

a clear degradation both in COMET and BLEU scores. Furthermore, it is noteworthy that Chat-GPT's sensitivity to the temperature varies depending on the language pair: the impact of temperature is relatively small when translating to high-resource languages, e.g., German, while for complex languages, e.g., Chinese, it has a large degradation in performance ($-4.3$ COEMT points and $-3.7$ BLEU points for Chinese) when the temperature changes from 0 to 1. We speculate that the huge resource variance in training data leads to differences in the confidence of languages, which partially explains the different performances. In the following experiments, we adopt $T = 0$ as our default setting to make the most of ChatGPT and ensure the stability of generation to avoid a result of noise.

| System | COMET | BLEU | ChrF | COMET | BLEU | ChrF |
|---|---|---|---|---|---|---|
| | DE⇒EN | | | EN⇒DE | | |
| Google Translator | **77.7** | **47.4** | **70.5** | **70.5** | **44.4** | **68.9** |
| ChatGPT | 77.2 | 43.5 | 69.4 | 69.3 | 40.6 | 67.1 |
| ChatGPT + TSP | 77.5† | 44.1 | 69.7 | 69.4 | 40.4 | 67.0 |
| | ZH⇒EN | | | EN⇒ZH | | |
| Google Translator | **73.5** | **33.5** | **61.2** | **68.5** | **48.8** | **43.8** |
| ChatGPT | 71.3 | 26.4 | 58.3 | 66.4 | 45.1 | 39.0 |
| ChatGPT + TSP | 71.5 | 26.7 | 58.4 | 67.2† | 45.3 | 39.3 |
| | RO⇒EN | | | EN⇒RO | | |
| Google Translator | **82.4** | **48.0** | **71.2** | 91.6 | **43.3** | **67.0** |
| ChatGPT | 80.6 | 41.8 | 68.8 | 92.4 | 40.6 | 65.5 |
| ChatGPT + TSP | 80.8 | 41.9 | 69.0 | **92.9†** | 40.8 | 65.7 |
| | ZH⇒RO | | | RO⇒ZH | | |
| Google Translator | 73.9 | **25.8** | **53.9** | 62.3 | **42.3** | **37.8** |
| ChatGPT | 73.8 | 20.9 | 51.5 | 58.9 | 37.7 | 33.3 |
| ChatGPT + TSP | **74.1†** | 21.0 | 51.3 | 59.1† | 38.0 | 33.7 |

Table 3: Performance with different prompts on 4 language pairs from Flores-200. "TSP" denotes our proposed task-specific prompting method. The best scores across different systems are marked in **bold** and the best scores of ChatGPT are underlined. Notably, we set the temperature as 0 for ChatGPT in this experiment. We can see that our TSP method consistently boosts the performance of ChatGPT in most settings. Shadowed areas mean difficult English-centric translation tasks, Green areas mean non English-centric translation tasks. "†" indicates a statistically significant difference from the ChatGPT baseline ($p < 0.05$).

## 3.2 The Effect of Task Information 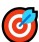

Previous studies (Jiao et al., 2023; Hendy et al., 2023) have shown that ChatGPT can achieve exceptional performance in conversational domain translation, which is attributed to its ability to generate more natural and diverse spoken language. However, given that ChatGPT is deliberately designed as a general task solver (Qin et al., 2023), when asking the ChatGPT to perform as a specific task engine, there will arise a task gap. This task inconsistency may limit ChatGPT's effectiveness in translation tasks other than the spoken domain.

To bridge the task gap and generate more translation-like sentences, we propose *Task-Specific Prompts* (TSP) to emphasize the translation task information. Specifically, we prepend the sentence *"You are a machine translation system."* to the best translation template in Jiao et al. (2023), and adopt it to query ChatGPT. The templates of prompts present in Table 2, and [TGT] represents the target languages of translation.

We have compared the performance of various

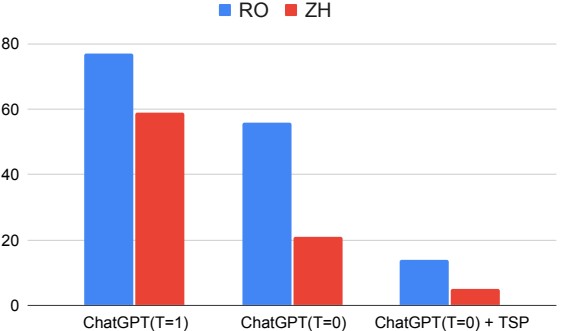

Figure 3: Number of Post-Edited sentences in non-English-centric language pairs, where a higher value means the translation contains more hallucinations. RO represents the translation for ZH⇒RO, while ZH represents the translation for ZH⇒RO.

models on four language pairs, covering eight distinct translation directions. These languages comprise 1) German, which is one of the most non-English languages in the GPT training data, 2) Romanian, a less frequently encountered non-English language in the GPT training data, and 3) Chinese, a large-scale language with a script distinct from

English. We also adopt Chinese-Romanian as a non-English-centric use case. Table 3 lists the full results, where we list both English-centric and non-English-centric language directions (marked with green ), and also, among English-centric directions, we highlight the difficult pairs (EN-ZH and EN-RO with shadow ) in terms of their resources and language distance.

### 3.2.1 English-Centric Language Pairs

We first consider the performance of Chat-GPT in English-centric translation language pairs. Specifically, we conduct experiments in three language pairs: German⇔English (high-resource), Romanian⇔English (low-resource), and Chinese⇔English (distant language).

**Results.** Our results presented in Table 3 show that our TSP method achieves comparable results on COMET score compared to Google Translator and even outperforms it in some language pairs, e.g., English⇒Romanian (92.9 v.s. 91.6). We also observe that our TSP method consistently improves the performance of vanilla ChatGPT, especially when translating to low-resource or distant languages. Specifically, our TSP method brings +0.8 and +0.5 COMET score improvements in English⇒Chinese and English⇒Romanian, respectively, and +0.2 on average when translating to English. We speculate that the high-resource training data can help the model better understand the specific task from a few task-related navigations, thereby reducing the need for additional task-specific information. Although our proposed TSP consistently improves the performance in terms of semantic metric, i.e., COMTE, notably, we have not consistently bridged the task gap in terms of lexical metrics (BLEU and ChrF), which is consistent with similar findings from Vilar et al. (2022) on PALM-540B model.

### 3.2.2 Non-English-Centric Language Pairs

We also evaluate the performance of ChatGPT in non-English-centric language pairs (since the pre-training process was dominated by the English tokens and the multilingual MT community argues it may harm the non-English-centric performance (Costa-jussà et al., 2022; Zan et al., 2022a, 2023).). We have an important finding that, 🚨 **when tackling non-English-centric MT language pairs, ChatGPT tends to generate translation hallucinations**, that is, some unrelated in-

| Target Language | Template |
|---|---|
| **Chinese** | `[Ro] would be translated to: [Zh];` `[Zh] (Note: ...)` |
| **Romanian** | `[Zh] can be translated into Romanian` `as [Ro]; [Ro] (Note: ...)` |

Table 4: Some templates about irrelevant information in generated sentences for Chinese⇔Romanian. Semi-colon is used to separate different templates. [Ro] represents the sentence in Romanian while [Zh] represents that in Chinese.

formation obeyed some patterns followed the translation, such as *"Translation may vary depending on context"*, which will greatly affect the MT performance. We used a post-processing method to remove irrelevant information from the generated text. Specifically, we summarize some templates about irrelevant sentences and remove them from the generation texts. Some templates are shown in Table 4 and the number of post-processed sentences is presented in Figure 3.

**Results.** Figure 3 shows that lower temperature can reduce the number of hallucinations (especially in distant languages, e.g., Chinese) and our TSP method can further reduce its number, which suggests that our method can help ChatGPT to better serve as a machine translation system. The full results on Romanian⇔Chinese lists are in Table 3. As seen, our TSP method can only slightly improve ChatGPT's performance, which could be due to the difficulty in both understanding and generating the language pairs. Meanwhile, our used post-editing approach could only roughly remove the hallucination patterns, ***the NLP/MT community should pay more attention to the potential hallucination when using ChatGPT to tackle the non-English text***.

The subsequent experiments will use ChatGPT with TSP as the default setting.

### 3.3 The Effect of Domain Information 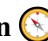

Compared with traditional machine translation systems, ChatGPT can incorporate additional information through the prompts to further improve its performance. While previous studies have shown that ChatGPT has great robust translation capabilities (Hendy et al., 2023), we believe that we can further enhance its performance by incorporating domain-specific guidance.

To this end, we propose Domain-Specific Prompts (DSP) that identify the domain informa-

| Method | Translation Prompt |
|---|---|
| **ChatGPT** | `"role": "system", "content": "You are a machine translation system.", "role": "user", "content": 'Please provide the [TGT] translation for the following sentence: '` |
| **ChatGPT+DSP** | `"role": "system", "content": "You are a machine translation system that translates sentences in the [DOM] domain.", "role": "user", "content": 'Please provide the [TGT] translation for the following sentence: '` |
| **ChatGPT+F-DSP** | `"role": "system", "content": "You are a machine translation system that translates sentences in the [FDOM] domain.", "role": "user", "content": 'Please provide the [TGT] translation for the following sentence: '` |

Table 5: Domain-Specific translation prompts. "`[DOM]`" and "`[FDOM]`" denote the correct and incorrect domain instructions, respectively.

tion of translated sentences in prompts to facilitate ChatGPT's generalization. Specifically, we ask ChatGPT with the following prompts *"You are a machine translation system that translates sentences in the [DOM] domain"*, as shown in Table 5. Here, `[DOM]` represents the correct domain of the translated sentence, while `[FDOM]` represents the wrong domain of that, which is used to verify whether the improvement comes from domain information. For example, for a biomedical sentence, `[DOM]` is *biomedical*, while `[FDOM]` can be any field except *biomedical*.

We evaluate our method on the WMT19 Bio and News datasets followed Jiao et al. (2023), which allows us to examine domain bias's impact. For example, the WMT19 Bio test set comprises Medline abstracts that require domain-specific knowledge, while the WMT19 News dataset features news-style texts that are significantly different from dialogues. To further prove the effectiveness of our method, we conduct our method on WMT22 English-Chinese E-Commerce test set, which is less likely to overlap with the GPT training data.

**Results.** The results are listed in Table 6. Obviously, the original ChatGPT does not perform as well as Google Translator in both COMET and lexical metrics (e.g., BLEU). However, our DSP method can consistently improve the performance of ChatGPT in terms of COMET score and even outperforms Google Translator in two datasets (WMT19 Bio Chinese ⇒ English and WMT19 News English ⇒ Chinese). This finding indicates

that our method can further improve the generalization ability of ChatGPT and narrow the gap with one of the most advanced commercial systems – Google Translator. Nonetheless, our method's impact on BLEU is inconsistent, and it still lags significantly behind Google Translator's performance.

To verify that the observed improvement is indeed due to the introduction of the domain information, we deliberately provided incorrect domain information for each sentence, namely *F-DSP*, to attack the improvement brought by the DSP strategy. Specifically, We exchange domain information for the biomedical sentences and the news sentences. We expect that the wrong domain guidance (F-DSP) will under-perform the DSP, and even perform worse than the vanilla ChatGPT. The results of these experiments are shown in the last row of Table 6, which clearly shows a consistent degradation in COMET, proving that the domain information is the key to the success of our method.

All the above DSP and F-DSP results confirm the importance of domain-specific prompting guidance in using ChatGPT for MT tasks.

## 4 Few-shot Machine Translation

In this section, we simply explore the effects of advanced in-context learning (ICL) strategies, specifically, we investigate ChatGPT's few-shot ICL and Chain-of-Thought (CoT) abilities on MT tasks.

### 4.1 Few-Shot In-Context learning

In-context learning (Brown et al., 2020b) has shown its remarkable ability for many NLP tasks (Liu et al., 2023). To further explore the capabilities of the ChatGPT, we conduct experiments with different sample selection strategies. Specifically, we evaluate the performance of few-shot machine translation in the following three directions: English⇒Chinese, English⇒Romanian, and English⇒German in Flores-200. We conducted experiments with randomly and TopK (Liu et al., 2022) sampled demonstrations from development sets in the 1-shot and 3-shot settings.

**Results.** Our results are listed in Table 7. As seen, in-context learning with random examples consistently improves the performance in both lexical metric (BLEU) and COMET score compared to the zero-shot approach, and increasing the number of shots can lead to further improvement, which is consistent with previous finding (Hendy et al., 2023). The advanced sample-selection strategy like

| System | WMT19 Bio | | | | WMT19 News | | | | WMT22 E-Commerce | |
|---|---|---|---|---|---|---|---|---|---|---|
| | EN⇒ZH | | ZH⇒EN | | EN⇒ZH | | EN⇒DE | | EN⇒ZH | |
| | COMET | BLEU | COMET | BLEU | COMET | BLEU | COMET | BLEU | COMET | BLEU |
| Google Translator | **59.4** | **38.8** | 59.4 | **36.1** | 59.3 | **43.4** | **64.1** | **33.7** | **71.7** | **48.0** |
| ChatGPT | 58.6 | 35.5 | 58.7 | 31.1 | 58.8 | 39.6 | 63.1 | 31.3 | 68.2 | 43.5 |
| ChatGPT + DSP | 58.9 | 35.8 | 59.6† | 31.3 | 59.6† | 39.8 | 63.2 | 31.5 | 68.6 | 43.8 |
| ChatGPT + F-DSP | 58.6 | 35.6 | 58.4 | 31.3 | 57.9 | 39.0 | 62.0 | 31.2 | 67.1 | 43.3 |

Table 6: Performance of ChatGPT on translation robustness, i.e., different domains. "DSP" denotes our proposed domain-specific prompting method, while "F-DSP" denotes the false domain-specific prompting, i.e., we specify wrong/unrelated domain information in the prompt. The results in green denote that "DSP" improves ChatGPT by a clear margin (0.5 (↑) score), while the red results denote the significant performance drops caused by "F-DSP". "†" indicates a statistically significant difference from the ChatGPT baseline ($p < 0.05$).

| System | EN ⇒ DE | | EN ⇒ ZH | | EN ⇒ RO | |
|---|---|---|---|---|---|---|
| | COMET | BLEU | COMET | BLEU | COMET | BLEU |
| Google Translator | **70.5** | **44.4** | 68.5 | **48.8** | 91.6 | **43.3** |
| ChatGPT | 69.4 | 40.4 | 67.2 | 45.3 | 92.9 | 40.8 |
| *Random Sampling few-shot prompting* | | | | | | |
| -w/ 1-shot | 69.8 | 40.6 | 67.6 | 45.4 | 93.1 | 40.7 |
| -w/ 3-shot | 70.0 | 40.7 | 68.3 | 45.9 | 93.6 | 40.9 |
| *TopK Sampling few-shot prompting* | | | | | | |
| -w/ 1-shot | 69.8 | 41.0 | 68.4 | 45.8 | 93.1 | 40.7 |
| -w/ 3-shot | **70.5** | 40.9 | **68.8** | 45.8 | **94.0** | 41.2 |

Table 7: Few-shot translation performance of ChatGPT on Flores-200. In the random sampling few-shot prompting setting, we randomly sample 1/3 examples from the development set with 3 runs. The best scores across different systems are marked in **bold** and the best scores of ChatGPT are underlined.

TopK, which chooses test-sample similar examples as demonstrations, can further improve the performance, even outperform Google Translator in some language pairs, e.g., English⇒Romanian (94.0 v.s. 91.6) and English⇒Chinese (68.8 v.s. 68.5).

We encouragingly find that the advanced sample-selection strategy for in-context learning for MT tasks upon ChatGPT is extremely similar to the design philosophy of example-based machine translation (EBMT, Nagao, 1984), where the EBMT is often characterized by its use of a bilingual corpus as its main knowledge base, at run-time. It is worthy of designing better ICL strategies inspired by EBMT in future work.

## 4.2 Chain-of-Thought

Chain-of-Thought (CoT) prompting (Wei et al., 2022c) has been demonstrated to be effective in eliciting the reasoning ability of large language models. Previous studies have shown that CoT can improve the ChatGPT's performance in natural language understanding tasks (Zhong et al., 2023), but **its influence on machine translation tasks has hardly been investigated**.

To investigate this further, we randomly select 20 samples from the test set and adopt the zero-shot CoT technique (Kojima et al., 2022) and the 1-shot CoT technique. Specifically, as shown in Table 8, for zero-shot CoT, we use the prompt `"Please provide the [TGT] translation for the following sentence step by step"` to extract step-by-step translation. We also add the sentence `'and then provide the complete sentence:'` to the end of the prompting to ensure that ChatGPT can generate the complete translation. While for the 1-shot CoT, we provide the manual intermediate reasoning steps inspired by zero-shot CoT, as shown in Table 8. Here, [S] and [T] represent the corresponding source and target sentence in the demonstration, respectively, and [S_i] and [T_i] are the i-th matching tokens in the source and target sentence.

**Results.** We conduct experiments in the following two translation directions: English⇒German

| Method | Translation Prompt |
|---|---|
| **Zero-Shot CoT** | `"role": "system", "content": "You are a machine translation system.", "role": "user", "content": 'Please provide the German translation for the following sentence step by step and then provide the complete sentence: '` |
| **1-Shot CoT** | `"role": "system", "content": "You are a machine translation system.", "role": "user", "content": 'Please provide the German translation for the following sentence step by step and then provide the complete sentence: [S] 1. [S_1] - [T_1] 2. [S_2] - [T_2] ... n. [S_n] - [T_n] The complete sentence in [TGT] is: [T] Please provide the German translation for the following sentence step by step and then provide the complete sentence:'` |

Table 8: The templates of Zero-Shot CoT and 1-shot CoT. [S_n] represents the $n$-th token in source demonstration [S], [T_n] represents the $n$-th token in target demonstration [T].

| Method | EN⇒DE | | EN⇒ZH | |
|---|---|---|---|---|
| | **COMET** | **BLEU** | **COMET** | **BLEU** |
| ChatGPT | 72.4 | 36.5 | 68.3 | 41.4 |
| -w zero-shot CoT | 69.3 (↓3.1) | 35.1 (↓1.4) | 59.5 (↓8.8) | 36.2 (↓5.2) |
| -w 1-shot CoT | 69.6 (↓2.8) | 37.0 (↑0.5) | 61.1 (↓7.2) | 37.6 (↓3.8) |

Table 9: Performance of ChatGPT equipped with CoT prompting methods on randomly selected 20 samples from English⇒German and English⇒Chinese.

and English⇒Chinese. The results are listed in Table 9, which shows that there is a significant degradation in COMET score with zero-shot CoT setting, especially in English⇒Chinese, which drops 8.8 COMET points. 1-shot CoT prompting can consistently outperform zero-shot CoT but still lags behind zero-shot prompting on COMET.

We looked in detail at the sentences generated by different prompts, presented in Table 10, and we have a negative but interesting observation: ***the CoT prompt leads to word-by-word translation behavior, which is the main reason for the significant translation degradation***.

For more CoT variants designed with different principles inspired by the philosophy in statistical MT (Zens et al., 2002; Koehn, 2009) will be explored in the future. For example, word-by-word and then reordering the translation (Du and Way, 2017; Ding et al., 2020), phrase-to-phrase (Feng et al., 2018; Ding et al., 2021) and then reordering the translation, and structure-to-structure transla-

tion (Kaplan et al., 1989).

# 5  Related Work

**Large Language Models.**  Large language models (LLMs) usually refer to language models with hundreds of billions of parameters, which are trained on massive text data (Zhao et al., 2023). LLMs usually can be classified into three groups based on model architectures: 1) encoder-only LLMs (Devlin et al., 2019; Liu et al., 2019; Zhong et al., 2022), usually used for NLU tasks; 2) decoder-only LLMs (Radford et al., 2019; Brown et al., 2020a), more suitable for NLG tasks; and 3) encoder-decoder LLMs (Raffel et al., 2020; Lewis et al., 2020; Zan et al., 2022b; Peng et al., 2023), which can achieve better performance on conditional text generation tasks.

Traditionally, these PLMs can achieve remarkable performance in various natural language processing (NLP) tasks through fine-tuning on specific tasks. But with the scaling up and the development of LLMs (Brown et al., 2020a; Ouyang et al., 2022b), decoder-only LLMs exhibit remarkable zero-shot and few-shot abilities, denoted emergent abilities (Wei et al., 2022b), and achieve comparable results with other LLMs in NLU and conditional NLG tasks. Especially the emergency of ChatGPT, developed by OpenAI, takes LLMs a big step forward in both academia and industry. ChatGPT possesses diverse abilities of NLP and can generate human-like responses by instruction-tuning (Wei et al., 2022a) and Reinforcement Learning from Human Feedback (RLHF) technique (Ouyang et al., 2022b).

**ChatGPT for Machine Translation.**  The ability of ChatGPT has been widely studied in various domains (Qin et al., 2023; Zhong et al., 2023), but its ability on machine translation tasks has not been fully investigated.  Jiao et al. (2023) and Hendy et al. (2023) first provided an evaluation on the performance of ChatGPT for machine translation, they found that ChatGPT can perform competitively with commercial translation products on high-resource European languages but lags behind significantly on low resource or distant languages. However, they usually adopt simple prompts and basic settings which cannot fully exploit the capabilities of ChatGPT, we first proposed that ChatGPT can achieve comparable results with proper settings and investigate how to make the most of ChatGPT for machine translation.

Subsequent work follows our work to further explore the performance of ChatGPT, Gao et al. (2023) and Lu et al. (2023a) introduce new information (e.g., POS or multilingual dictionaries), He et al. (2023) proposed a CoT-like framework to generation human-like translation.

## 6 Conclusion

In this paper, we investigate how to further mine ChatGPT's translation ability from three perspectives, namely temperature, task, and domain information, and correspondingly propose an optimal temperature setting and two simple but effective prompts. We empirically demonstrated that there is a high correlation between temperature and ChatGPT's performance, and a lower temperature usually can achieve better performance. Experimental results across various language pairs and domains proved the effectiveness of our proposed prompts. We further explore the effectiveness of advanced in-context learning strategies for Chat-GPT, we find that the few-shot in-context learning method can consistently improve ChatGPT's performance, while conventional Chain-of-Thought (CoT) prompting will degrade its performance because of its word-by-word translation behavior.

In future work, besides the aforementioned explorations (EBMT-inspired prompts designing, statistical MT-inspired chain-of-thought designing), we would like to investigate how to further elicit the ability of ChatGPT by designing more effective prompts (e.g., design human-like CoT to navigate the LLMs, and better demonstration selection algorithms in few-shot ICL) and investigate the ability of ChatGPT for more MT settings (e.g., document translation).

## Limitations

Our work has several potential limitations. First, we only propose some simple prompts that have not been carefully designed to investigate the capabilities of ChatGPT, which may not sufficiently elicit the power of ChatGPT. Second, we have not fully studied the performance of ChatGPT in few-shot scenarios, especially the effect of Chain-Of-Thought in machine translation. In future work, we would like to design different types of prompts to further improve ChatGPT's performance in machine translation and conduct more in-depth analyses and discussions.

## Ethics Statement

We take ethical considerations very seriously and strictly adhere to the EMNLP Ethics Policy. This paper focuses on exploring the translation ability of ChatGPT on open-sourced machine translation datasets, not involving any ethics problem. Both the compared models and evaluation datasets used in this paper are publicly available and have been widely adopted by researchers. Therefore, we believe that this research will not pose ethical issues.

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

---

*Zero-shot prompting*

**System:** You are a machine translation system.

**User: Please provide the German translation for the following sentence:** Sadly, as newer methods of writing have emerged, the pencil has been relegated to lesser status and uses.

**Response:** Leider wurde der Bleistift durch neuere Schreibmethoden in eine geringere Position und Verwendung gedrängt.

---

*Zero-shot CoT prompting*

**System:** You are a machine translation system.

**User: Please provide the German translation for the following sentence step by step and then provide the complete sentence:** Sadly, as newer methods of writing have emerged, the pencil has been relegated to lesser status and uses.

**Response:** 1. Sadly - Bedauerlicherweise 2. as newer methods of writing have emerged - da neuere Schreibmethoden aufgetaucht sind 3. the pencil - der Bleistift 4. has been relegated - wurde zurückgestuft 5. to lesser status and uses - zu geringerem Status und Verwendungszwecken Bedauerlicherweise wurde der Bleistift aufgrund neuerer Schreibmethoden zu geringerem Status und Verwendungszwecken zurückgestuft.

---

*1-shot CoT prompting*

**System:** You are a machine translation system.

**User:** Please provide the German translation for the following sentence step by step and then provide the complete sentence: That said, expect to be out of breath, and take care in the steeper portions, especially when wet, as it can become dangerous quickly. 1. That said - jedoch damit 2. expect - Rechnen sie 3. to be - zu sein 4. out of breath - außer Atem 5. and - und 6. take care - vorsichtig 7. in the steeper portions - in den steileren Abschnitten 8. especially - besonders 9. when wet - wenn es nass ist 10. become - werden 11. dangerous - gefährlich 12. quickly - schnell The complete sentence in German is: Rechnen Sie jedoch damit, außer Atem zu sein, und seien Sie in den steileren Abschnitten vorsichtig, besonders wenn es nass ist, da es dort schnell gefährlich werden kann. **Please provide the German translation for the following sentence step by step and then provide the complete sentence:** Sadly, as newer methods of writing have emerged, the pencil has been relegated to lesser status and uses.

**Response:** 1. Sadly - Leider 2. as - da 3. newer methods - neuere Methoden 4. of writing - des Schreibens 5. have emerged - aufgetaucht sind 6. the pencil - der Bleistift 7. has been relegated - wurde zurückgestuft 8. to lesser status - zu einem geringeren Status 9. and uses - und Verwendungen. The complete sentence in German is: Leider wurde der Bleistift aufgrund des Auftauchens neuerer Methoden des Schreibens zu einem geringeren Status und Verwendungen zurückgestuft.

---

Table 10: Examples of zero-shot and 1-shot CoT prompting on English⇒German. The sentences in red are the reasoning step for CoT and those in blue are the one-shot example. Sentences in **bold** are the instruction of CoT.