# OpenReview forum: "Towards Making the Most of ChatGPT for Machine Translation"
_EMNLP/2023/Conference — EMNLP 2023 Findings_

### Official Review · Reviewer_4hHT · 2023-07-28

**Typos Grammar Style And Presentation Improvements:** No
**Soundness:** 4

**Excitement:**

4: Strong: This paper deepens the understanding of some phenomenon or lowers the barriers to an existing research direction.

**Missing References:**

No

**Paper Topic And Main Contributions:**

ChatGPT is widely adopted in various NLP tasks. However, the previous works on using ChatGPT for machine translation only adopt simple prompts and analyses. This paper conducts a more in-detailed study on using ChatGPT for MT. Specifically, the authors take temperature, task information as well as domain information into account and propose two novel prompting strategies. The main findings are: 1) ChatGPT’s performance largely depends on the temperatures and a lower temperature can result in higher performance; 2) Emphasizing the task information and domain information can improve the performance; 3) the translation between non-English pairs suffer from more hallucination.

Presentation: Good
This paper is well-written, with different colors and icons, making it easier for the reader to understand.
Soundness: Good
The evaluation is well and sufficient, with multiple datasets in different metrics.
Novelty: Good
This is the first paper that takes a deeper look at using ChatGPT for MT.
Significant: Fair
The improvement of the proposed two prompts is not significant.


**Questions For The Authors:**

1. Can you explain why lower temperatures can lead to better performance?
2. Can you explain why non-centric translation can lead to more hallucinations?
3. Just curiosity, what is the performance of GPT4?

**Reasons To Accept:**

1. This paper is well-written and easy to follow.
2. Interesting and timely topic, which can contribute to the machine translation field.
3. The evaluation is solid, with multiple datasets in different metrics.

**Reasons To Reject:**

1. Although it is acceptable for EMNLP, this paper is too empirical without any theoretical understanding.
2. The improvement of the two proposed prompts is not significant.

**Reproducibility:**

3: Could reproduce the results with some difficulty. The settings of parameters are underspecified or subjectively determined; the training/evaluation data are not widely available.

**Reviewer Confidence:**

4: Quite sure. I tried to check the important points carefully. It's unlikely, though conceivable, that I missed something that should affect my ratings.

---

> ### Author Rebuttal · Authors · 2023-08-29
>
> Appreciate your positive comments. We will attend to all comments to the best extent, and try to address your concerns as follows:
>
> **Q1: “Too empirical without any theoretical understanding”**
>
> Thanks for your advice, we will consider adding some theoretical formulations from the perspective of explaining why prompting engineering and in-context learning works.
>
> **Q2: “The slight improvement”**
>
> We conducted the significance test (p<0.05) and showed that our TSP and DSP strategies bring significant improvement on 7/12 tasks, and we will add the significance test to the revised version.
>
> Also, our method can be further improved by adjusting more hyperparameters, such as temperature and top-k/p decoding, and our aim is to provide simple but effective guidance to enhance translation performance. In the revised version, we will try to optimize the performance according to the hyperparameter grid search.
>
> **Q3: “Why lower temperatures can lead to better performance”**
>
> As mentioned in section 3.1 and the introduction, previous work [1] has proved that decoding with higher temperatures displays greater linguistic variety, while the low one generates grammatically correct and deterministic text. For tasks with a high degree of certainty, such as translation, a lower temperature can generate more grammatically correct and deterministic text which can avoid stochasticity and achieve a stable and better performance.
>
> [1] Ippolito et al., Comparison of diverse decoding methods from conditional language models. In ACL 2019.
>
> **Q4: “Why non-centric translation can lead to more hallucinations”**
>
> We argued that it may be because ChatGPT is mainly trained by English corpus, which may limit its ability to understand and generate other languages, which will lead to more hallucinations.
>
> **Q5: “Performance of GPT4”**
>
> Thanks for pointing this out, given the significantly higher cost and inference latency (than GPT3.5), we only test our method in the WMT19 Bio En-Zh dataset with GPT-4. The results are shown below.
> The results show that our method can stably improve the performance of GPT-4.
>
> |        | BLEU |  COMET |
> |:-----------:|:----:|:----:|
> |  GPT-4  |  33.6  |  56.9 |
> |  +TSP  |  **34.0**  |  **58.0** |

---

### Official Review · Reviewer_ZQYG · 2023-08-04

**Soundness:** 2

**Excitement:**

2: Mediocre: This paper makes marginal contributions (vs non-contemporaneous work), so I would rather not see it in the conference.

**Paper Topic And Main Contributions:**

The authors conducted empirical experiments on the ChatGPT with the aim of enhancing its translation performance. These experiments were focused on the multilingual test sets Flores-200 and WMT19（News + Biomedical). The authors investigated both decoding temperature and translating prompts and obtained some empirical conclusions.

**Reasons To Accept:**

1. The paper is well-written and easy to follow.
2. The authors conducted experiments on multiple test sets (Multilingual / Multi-domain ) to confirm their findings.


**Reasons To Reject:**

1. The contributions and novelty of the work are limited. After reading the paper, I am trying to figure out the research question, which may be a better translation prompt. The paper is more like a product manual telling me how to use ChatGPT for translation.

2. The slight improvement （ ChatGPT vs. ChatGPT+DSP in Table 6 and Table 7）can not support the claim  “Experimental results across various language pairs and domains proved the effectiveness of our proposed prompts.”

**Reproducibility:**

4: Could mostly reproduce the results, but there may be some variation because of sample variance or minor variations in their interpretation of the protocol or method.

**Reviewer Confidence:**

5: Positive that my evaluation is correct. I read the paper very carefully and I am very familiar with related work.

---

> ### Author Rebuttal · Authors · 2023-08-29
>
> Thanks for your careful reading and detailed comments. We will attend to all comments to the best extent, and try to address your concerns as follows:
>
> **Q1: Contribution and Novelty of the work**
>
> Our initial motivation is to prove the powerful ability of ChatGPT in machine translation and provide several simple strategies to maximize its performance, which has been questioned in previous work, for example, Jiao et al. (2023) reported in their position paper (arXiv: 2301.08745v1) that ChatGPT lags behind significantly on low-resource/ distant languages and different domains like biomedical.
>
> In this work, we are (nearly the first) to find that ChatGPT’s translation performance largely depends on the temperatures, and emphasizing the task information and domain information can improve the performance. Through our experiments, we encouragingly show the MT and LLM community that it's not that ChatGPT is not effective in translation, but that we used it in an improper way.
>
> **Q2: “The slight improvement in Table 6 and Table 7”**
>
> First, we conducted the significance test (p<0.05) and showed that our TSP and DSP strategies bring significant improvement on 7/12 tasks.
>
> Also, given that the call for using model-based neural metrics, e.g., Comet, rather than BLEU score [1,2] and the evidence from previous ``translation with LLM’’ studies [3], improvement in terms of Comet is more convincing than BLEU, and in our experiments, we achieved up to 0.9 Comet improvement (on the WMT19 bio zh-en task).
>
> Moreover, our method can be further improved by adjusting more hyperparameters, such as temperature and top-k/p decoding, and our aim is to provide simple but effective guidance to enhance translation performance. Therefore, we believe our improvements are reasonable.
>
> [1] Matt Post, A Call for Clarity in Reporting BLEU Scores. In WMT 2018.
>
> [2] Freitag et al., Results of WMT22 Metrics Shared Task: Stop Using BLEU – Neural Metrics Are Better and More Robust. In WMT 2022.
>
> [3]  Hendy et al., How Good Are GPT Models at Machine Translation? A Comprehensive Evaluation. In arXiv preprint 2023.

---

### Official Review · Reviewer_gZrL · 2023-08-05

**Soundness:** 3

**Excitement:**

3: Ambivalent: It has merits (e.g., it reports state-of-the-art results, the idea is nice), but there are key weaknesses (e.g., it describes incremental work), and it can significantly benefit from another round of revision. However, I won't object to accepting it if my co-reviewers champion it.

**Missing References:**

That using more few-shot examples doesn't necessarily improve performance has been observed and discussed in the following papers:
[1] PaLM: Scaling Language Modeling with Pathways
[2] Prompting Large Language Model for Machine Translation: A Case Study

**Paper Topic And Main Contributions:**

This paper examines different prompting strategies for ChatGPT for machine translation tasks. The authors studied the impact of decoding temperature, task/domain-specific information, and few-shot and chain-of-thought prompting on translation performance with Flores and WMT benchmarks. The results offered some insights: 1) lower temperature benefits translation; 2) adding task and domain-specific information is helpful; 3) few-shot prompting improves translation but chain-of-thought doesn't work well.

**Questions For The Authors:**

1. Using WMT19 for evaluation raises concerns because ChatGPT might use WMT19 dataset for training, making the conclusion biased. Besides, the biomedical test set is small.
2. The authors conclude that "TSP benefits translation" but without whether task-general system prompt works, like "you are a professional system assistant and can help users with different tasks".
3. There are no significance tests for nearly all results in this paper. Considering that most improvements are modest, the gain might be a result of noise.
4. In line 384, the authors claim studying "different sample selection strategies" but they actually only experimented with random sampling. Did you also work with other selection methods? How many runs did you perform with random selection?
5. What does the "advanced sampling selection strategies" in line 403 mean?

While ChatGPT gains popularity, using it for scientific research should be more careful and rigorous. This is more concerned when only reporting its downstream performance. ChatGPT is a product. We all have no idea what data has been used for its training. Stating "Zero-Shot" as in Section 3 is not well justified.

**Reasons To Accept:**

This paper extensively experiments on prompting for ChatGPT for translation, which offers insights into the translation capability of ChatGPT.


**Reasons To Reject:**

* Experimental setup has problems.
* Some conclusions need further evidence.

**Reproducibility:**

4: Could mostly reproduce the results, but there may be some variation because of sample variance or minor variations in their interpretation of the protocol or method.

**Reviewer Confidence:**

4: Quite sure. I tried to check the important points carefully. It's unlikely, though conceivable, that I missed something that should affect my ratings.

---

> ### Author Rebuttal · Authors · 2023-08-29
>
> Thanks for your insightful comments, which will serve to improve our paper considerably. We will attend to all comments to the best extent, and try to address your concerns as follows:
>
> **Q1: “WMT19 makes the conclusion biased ... because ChatGPT might use dataset for training and biomedical test set is small”**
>
> Thanks for pointing this out, we have to explain that the aim of this study is to maximize the effectiveness of ChatGPT when functioning as a machine translator, therefore, we follow Jiao et al. (2023) (who is the first to show the possibility of translating with ChatGPT) to verify our proposed DSP on the WMT19 dataset.
>
> Although it is unclear what data has been used for ChatGPT training, we provide two pieces of evidence to dispel your concern about the convincing of our conclusion:
> 1) Table 6 shows that DSP can improve ChatGPT’s translation performance while FDSP degrades the translation performance, demonstrating the effectiveness of DSP;
> 2) We conduct experiments on a larger WMT22 English-Chinese e-commerce test set, which is less likely to overlap with the GPT training data, and report the results below. The extra experiments on the updated test data confirm the effectiveness of our proposed DSP.
>
> |        | BLEU |  COMET |
> |:-----------:|:----:|:----:|
> |  ChatGPT  |  43.5  |  68.2 |
> |  +DSP  |  **43.8**  |  **68.6** |
> |  +FDSP  |  43.3  |  67.1 |
>
> [1] Jiao et al., Is ChatGPT a good translator? a preliminary study. In arXiv preprint 2023.
>
> **Q2: “without .. task-general system prompt”**
>
> Given that ChatGPT is deliberately designed as a general task solver [2,3], we argue that ChatGPT is already the task-general system even without a task-general system prompt. To address your concern, we followed your suggestion and added the task-general system prompt “You are a professional system assistant.” in Flores-200 English-Chinese. The results below show that although ``+general prompt’’ slightly improves the COMET score, our TSP is still the best among them, showing the effectiveness of our method.
>
> |        | BLEU |  COMET |
> |:-----------:|:----:|:----:|
> |  ChatGPT  |  45.1  |  66.4 |
> |  +general prompt  |  45.1  |  66.8 |
> |  +TSP  |  **45.3**  |  **67.2** |
>
> [2] Qin et al., Is ChatGPT a General-Purpose Natural Language Processing Task Solver? In arXiv preprint 2023.
>
> [3] OpenAI, GPT-4 Technical Report. In OpenAI blog 2023.
>
> **Q3: “no significance tests … gain might be a result of noise”**
>
> We followed your suggestion and conducted the significance test (p<0.05) with comet-compare, and found that 1) TSP has a significant improvement over baseline on 5/8 tasks, and 2) DSP has a significant improvement over baseline on 2/4 tasks, illustrating the consistent effectiveness of our method. We will add the above significance tests to the revised version.
>
> As for the ``noise’’ concern, due to the zero temperature setting, all the better translation performance can be easily reproducible.
>
> **Q4: “Results of different sample selection strategies”**
>
> Thanks for pointing this out, in our current few-shot in-context learning (ICL) experiment (Table 7), we only report the results of the random sampling strategy in 1 run.
>
> To make the experiments more convincing, we adopted TopK strategies [4] and a random sampling strategy with 3 runs following the tasks in Table 7. The results are shown below. We can notice that 1) few-shot settings can consistently improve ChatGPT’s performance, and 2) the use of advanced few-shot sampling strategies, e.g., TopK, can further enhance its performance.
>
> For EN-> ZH:
> |        | BLEU |  COMET |
> |:-----------:|:----:|:----:|
> |  ChatGPT  |  45.3  |  67.2 |
> | *Random Sampling*  |
> |  -w/ 1-shot  |  45.4  |  67.6 |
> |  -w/ 3-shot  |  **45.9**  |  68.3 |
> | *TopK*  |
> |  -w/ 1-shot  |  45.8  |  68.4 |
> |  -w/ 3-shot  |  45.8  |  **68.8** |
>
> For EN-> DE:
> |        | BLEU |  COMET |
> |:-----------:|:----:|:----:|
> |  ChatGPT  |  40.4  |  69.4 |
> | *Random Sampling*  |
> |  -w/ 1-shot  |  40.6  |  69.8 |
> |  -w/ 3-shot  |  40.7  |  70.0 |
> | *TopK*  |
> |  -w/ 1-shot  |  **41.0**  |  69.8 |
> |  -w/ 3-shot  |  40.9  |  **70.5** |
>
> For EN-> RO:
> |        | BLEU |  COMET |
> |:-----------:|:----:|:----:|
> |  ChatGPT  |  40.8  |  92.9 |
> | *Random Sampling*  |
> |  -w/ 1-shot  |  40.7  |  93.1 |
> |  -w/ 3-shot  |  40.9  |  93.6 |
> | *TopK*  |
> |  -w/ 1-shot  |  40.7  |  93.1 |
> |  -w/ 3-shot  |  **41.2**  |  **94.0** |
>
> [4] Liu et al., What makes good in-context examples for GPT-3? In DeeLIO 2022.
>
> **Q5: "Zero-Shot" as in Section 3**
>
> In section 3, the "zero-shot’’ corresponds to the ``few-shot’’ in the in-context learning (ICL) scenario.
>
> **Q6: Missing References**
>
> Thanks for your advice, we promise to add these references in the revised version.

---

### Meta-Review · Area_Chair_DQko · 2023-09-23

**Recommendation:** 3

**Metareview:**

*Summary*: This work investigates different prompting strategies to improve ChatGPT's performance on machine translation tasks. The authors c
onsider the impact of decoding temperature, inserting task/domain-specific information to the prompts, and the effect of few-shot
and chain-of-thought prompting on translation quality on two (FLORES, WMT) benchmarks.

*Evaluation*: R1 and R3 have rated this work as 3 or higher on soundness (3/4), while R2 has rated this as a 2. R2's main concerns are that 1) the investigation in this work is too empirical without any other solid backing and 2) the content is a bit thin with the focus on identifying good prompting strategies to improve ChatGPT's translation capabilities. Regarding 1), well-conducted empirical analyses do have a place in EMNLP. With regards to 2), the authors have offered many natural dimensions (adding task/domain information, few-shot setting) along which prompting might be carried out for translation. The CoT-based prompting could have been made more competitive; it is not surprising that just adding "step by step" to the prompt does not help.

---

### Decision · Program_Chairs · 2023-10-07

**Decision:**

Accept-Findings

**Comment:**

*Summary*: This work investigates different prompting strategies to improve ChatGPT's performance on machine translation tasks. The authors c
onsider the impact of decoding temperature, inserting task/domain-specific information to the prompts, and the effect of few-shot
and chain-of-thought prompting on translation quality on two (FLORES, WMT) benchmarks.

*Evaluation*: R1 and R3 have rated this work as 3 or higher on soundness (3/4), while R2 has rated this as a 2. R2's main concerns are that 1) the investigation in this work is too empirical without any other solid backing and 2) the content is a bit thin with the focus on identifying good prompting strategies to improve ChatGPT's translation capabilities. Regarding 1), well-conducted empirical analyses do have a place in EMNLP. With regards to 2), the authors have offered many natural dimensions (adding task/domain information, few-shot setting) along which prompting might be carried out for translation. The CoT-based prompting could have been made more competitive; it is not surprising that just adding "step by step" to the prompt does not help.